# Plant-mediated community structure of spring-fed, coastal rivers

**Matthew V. Lauretta**[1]*, **William E. Pine III**[2], **Carl J. Walters**[3], **Thomas K. Frazer**[4]

**1** National Oceanic Atmospheric Administration, National Marine Fisheries Service, Miami, FL, United States of America, **2** Department of Wildlife Ecology and Conservation, University of Florida, Gainesville, FL, United States of America, **3** Fisheries Research Centre, University of British Columbia, Vancouver, British Columbia, Canada, **4** School of Natural Resources and Environment, University of Florida, Gainesville, FL, United States of America

* Matthew.Lauretta@noaa.gov

**Data Availability Statement:** All datasets, in their entirety, used in the analyses are provided as supporting information. The ecosystem model is also provided as supporting information for

## Abstract

Quantifying ecosystem-level processes that drive community structure and function is key to the development of effective environmental restoration and management programs. To assess the effects of large-scale aquatic vegetation loss on fish and invertebrate communities in Florida estuaries, we quantified and compared the food webs of two adjacent spring-fed rivers that flow into the Gulf of Mexico. We constructed a food web model using field-based estimates of community absolute biomass and trophic interactions of a highly productive vegetated river, and modeled long-term simulations of vascular plant decline coupled with seasonal production of filamentous macroalgae. We then compared ecosystem model predictions to observed community structure of the second river that has undergone extensive vegetative habitat loss, including extirpation of several vascular plant species. Alternative models incorporating bottom-up regulation (decreased primary production resulting from plant loss) versus coupled top-down effects (compensatory predator search efficiency) were ranked by total absolute error of model predictions compared to the empirical community observations. Our best model for predicting community responses to vascular plant loss incorporated coupled effects of decreased primary production (bottom-up), increased prey search efficiency of large-bodied fishes at low vascular plant density (top-down), and decreased prey search efficiency of small-bodied fishes with increased biomass of filamentous macroalgae (bottom-up). The results of this study indicate that the loss of vascular plants from the coastal river ecosystem may alter the food web structure and result in a net decline in the biomass of fishes. These results are highly relevant to ongoing landscape-level restoration programs intended to improve aesthetics and ecosystem function of coastal spring-fed rivers by highlighting how the structure of these communities can be regulated both by resource availability and consumption. Restoration programs will need to acknowledge and incorporate both to be successful.

replication of the study findings, or additional analysis by interested researchers.

**Funding:** The project was funded by the Florida Fish and Wildlife Service's State Wildlife Program (https://myfwc.com/conservation/special-initiatives/fwli/) which is supported by the U.S. federal State and Tribal Wildlife Grants Program.

**Competing interests:** The authors have declared that no competing interests exist.

## Introduction

Human alterations of the landscape and associated changes in the physical and chemical properties of aquatic environments have resulted in the loss of submersed vascular plants from shallow aquatic ecosystems around the world [1, 2, 3]. For example, in nutrient-enriched systems, algae capable of rapid nutrient uptake and growth can dominate and potentially exclude vascular plants with lower nutrient uptake, assimilation, and growth rates [4]. Because plants play a central role in the ecology of aquatic ecosystems in a variety of ways, including contributing to ecosystem production, modifying biogeochemical processes, and mediating biotic interactions [1, 5, 6, 7], changes in aquatic plant communities may have profound consequences on ecosystem structure. Rooted vascular plants link benthic substrates to the overlying water column through the uptake of nutrients from sediment pore waters and transport of organic matter, minerals, and gases to both the water and benthic environments [8, 9]. The above-ground structure provided by these plants also decreases sediment erosion and resuspension by reducing water velocity and turbulence at the water-sediment interface [8, 10, 11]. As a consequence, water clarity and light availability are increased and a positive feedback loop for vascular plant primary production is provided.

Plants directly support the production of higher trophic levels and have been shown to have a major influence on the abundance and diversity of stream faunal communities [12]. Plants mediate predator-prey interactions between fishes and invertebrates by providing refuge habitat for prey populations and decreasing prey encounter rates of predators [13], allowing predator and prey populations to coexist at relatively high densities compared to unstructured habitats [14, 15, 16]. The effects of vascular plant loss and replacement by algal species on the faunal communities that they support are not currently well-understood [17], but this replacement is observed globally and is increasingly of concern to water resource users and managers [2, 3, 18, 19].

Due to their unique hydrogeological and biological properties, Florida's spring-fed rivers are excellent model ecosystems to study the effects of vegetative habitat loss on community structure. These rivers generally have stable streamflow and water temperatures, as well as extremely high rates of primary production [7, 20] that support diverse communities of plants, invertebrates, and fishes [21, 22]. Spring systems are also currently of significant conservation concern because of increased demands on groundwater for consumptive uses, which can lead to reduced spring discharge [23, 24], salinization [23], altered nutrient dynamics [25], and changes in floral and faunal communities [18, 26]. Within these spring ecosystems, documented shifts in the composition and biomass of primary producers are temporally concordant with changes in watershed land-use, including increased agricultural and residential development, and associated changes in streamflow and water quality [25, 27]. Of particular concern is the rapid decline of once dominant vascular plants, including *Vallisneria americana*, *Potamogeton* spp., and *Sagittaria kurziana* in several systems, and the widespread proliferation of filamentous macroalgae, including *Chaetomorpha* spp., *Gracilaria* spp., and *Lyngbya* spp. [27, 28, 29], which may have effects on higher trophic levels [17].

The causal mechanisms of vascular plant loss from the rivers or the effects on faunal communities within these systems are not fully understood [18]. However, resource managers are tasked with developing restoration and management plans for these systems, creating a need for new science to inform restoration strategies. As an example, the loss of vascular plants, which provide forage and refuge habitat, may alter invertebrate grazer communities, predator-prey dynamics of fishes, and other important population-level interactions. Such alterations may lead to undesirable shifts in fish and invertebrate communities and loss of ecosystem

function, which is of critical importance to managers and resource users concerned with the conservation of these unique coastal ecosystems [30, 31, 32].

To evaluate fish and invertebrate community responses to vegetative habitat loss, we (1) conducted a large-scale field assessment of ecosystem structure and function in two adjacent spring-fed rivers (Chassahowitzka and Homosassa rivers) where changes in vascular plant biomass (and replacement by filamentous macroalgae) have occurred with different trajectories; (2) used this information to construct a food web model using empirical estimates of plant, invertebrate, and fish absolute biomass and trophic interactions from the more vegetated river (Chassahowitzka River); (3) simulated seasonal filamentous macroalgae blooms and vascular plant decline over long time periods to quantify the corresponding responses in fishes and invertebrates under alternative scenarios of primary producer biomass with coupled predator-prey dynamics (i.e., high algae or plant biomass resulting in low predator search efficiency and vice-versa); and, (4) evaluated the absolute error in predicted community structure by comparing model projections with the observed plant, invertebrate and fish absolute biomass and trophic dynamics from the adjacent river where native vascular plants have been largely extirpated and replaced by filamentous macroalgae (Homosassa River). Our fundamental questions are: "What are the overall effects of primary producer composition and biomass change on fish and invertebrate community structure in spring-fed, coastal rivers?" and "Which predator-prey trophic dynamics–specifically prey availability and predator search efficiency under high or low levels of either vascular plants or filamentous macroalgae–best explain the observed differences in community structure?" Our overall goal was to capture the predominant trophic dynamics between plants, invertebrates, and fishes that were best reflected in the community structures observed in the two river systems, and to develop an ecosystem model incorporating mechanistic dynamics that account for these observed patterns to guide restoration strategy evaluation.

## Materials and methods

### Study systems

The Chassahowitzka and Homosassa rivers are located along the west coast of peninsular Florida, north of Tampa Bay. Both rivers originate as large artesian springs with mean discharge >2.49 m³/s [33] and flow west directly into the Gulf of Mexico. Riparian zones differ between the two rivers; the Homosassa River has extensive residential development whereas the Chassahowitzka River is primarily within the confines of state and federal protected lands with residential development restricted to areas outside of park boundaries and primarily along canals near the headspring area. Clear spring water, stable sand and mud substrates, and relatively low stream flow (mean velocity <0.49 m/s) characterize both rivers [27, 34]. The abundance and diversity of rooted vascular plants has declined in both rivers over the last two decades [27, 35]; however, the rate of loss has differed between the two systems (Fig 1). The Chassahowitzka River is comprised of a relatively diverse assemblage of submersed aquatic plants compared to the Homosassa River, where open sand and mud substrate predominates [17, 27]. Filamentous macroalgae blooms occur during winter and spring in both systems, creating seasonal differences in vegetative habitat characteristics (Fig 2) [35]. Both rivers historically supported rich and diverse fish communities [17, 20, 35, 36].

### Estimation of biomass and diet composition from field surveys

Licensed state biologists with the Florida Fish and Wildlife Commission served as project field operation leads during the sampling of collection of fishes. The research was conducted under University of Florida Project Number 00069469 and Florida Fish and Wildlife Commission

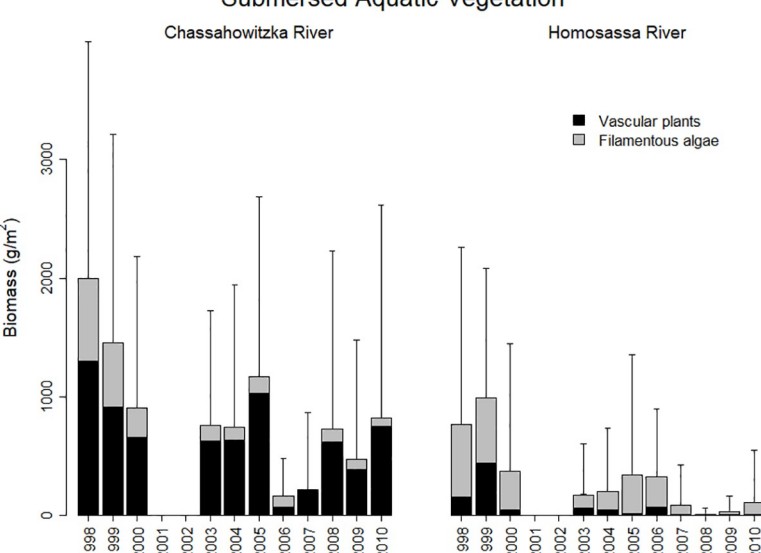

**Fig 1. Long-term mean vegetation biomass estimates at the Chassahowitzka and Homosassa rivers, Florida.** Note that sampling did not occur during 2001 and 2002.

(Commission) Agreement Number 07002. All field research was conducted in accordance with sampling permit requirement and standard agency protocols issued by the Commission and specific to the project. All sacrificed fishes were euthanized by hypothermic shock. Field sampling occurred over a three-year period from July 2007 to June 2010. Three study reaches were sampled in each river, representing a gradient from freshwater to brackish habitats. Table 1 lists the geographic coordinates of the upstream and downstream boundaries for each river reach. The reaches were located directly below the freshwater source springs (Reach 1), midway between the source springs and the salt marsh (Reach 2), and directly above the salt marsh (Reach 3). Submersed aquatic vegetation, macroinvertebrates, and small-bodied fishes were sampled during February and August in the first two years, and monthly in the third year [35]. Large-bodied fishes were sampled during January and July in the first two years using standardized mark-recapture electrofishing, and monthly in the third year [35, 36]. Small-bodied fishes were sampled by block net seine depletions [35, 36]. Macroinvertebrates were sampled by throw trap depletions [17], mesh sleeve grab sampling of aquatic vegetation, and by sediment coring. Aquatic vegetation was sampled by quadrat sampling at fixed, long term monitoring transects [35]. Large-bodied fish, small-bodied fish, and large crustacean catch data were corrected for incomplete detection and effort to estimate absolute abundances as described in companion papers focused on sampling methods [36, 37]. Absolute biomasses of fishes and invertebrates were estimated by multiplying the estimated abundances by the mean weight of individuals per study reach and year. Mean trophic group biomasses were estimated by a simple bootstrap (with replacement) of absolute biomass estimates across study reaches and years. Trophic group definitions of producers and consumers sampled from each of the river systems are listed in Table 2 and sampling methods used to estimate biomasses and diet compositions are summarized in Table 3 (see also [35, 36, 37] for additional details).

Trophic linkages were identified through analyses of diets from fish collected from 2007 to 2010. Large-bodied fishes were sampled for stomach contents by gastric lavage in the field, with exception of lake chubsucker (*Erimyzon sucetta*) and American eel (*Anguilla rostrata)*, which were sacrificed and examined by dissection in the lab. All small-bodied fishes examined

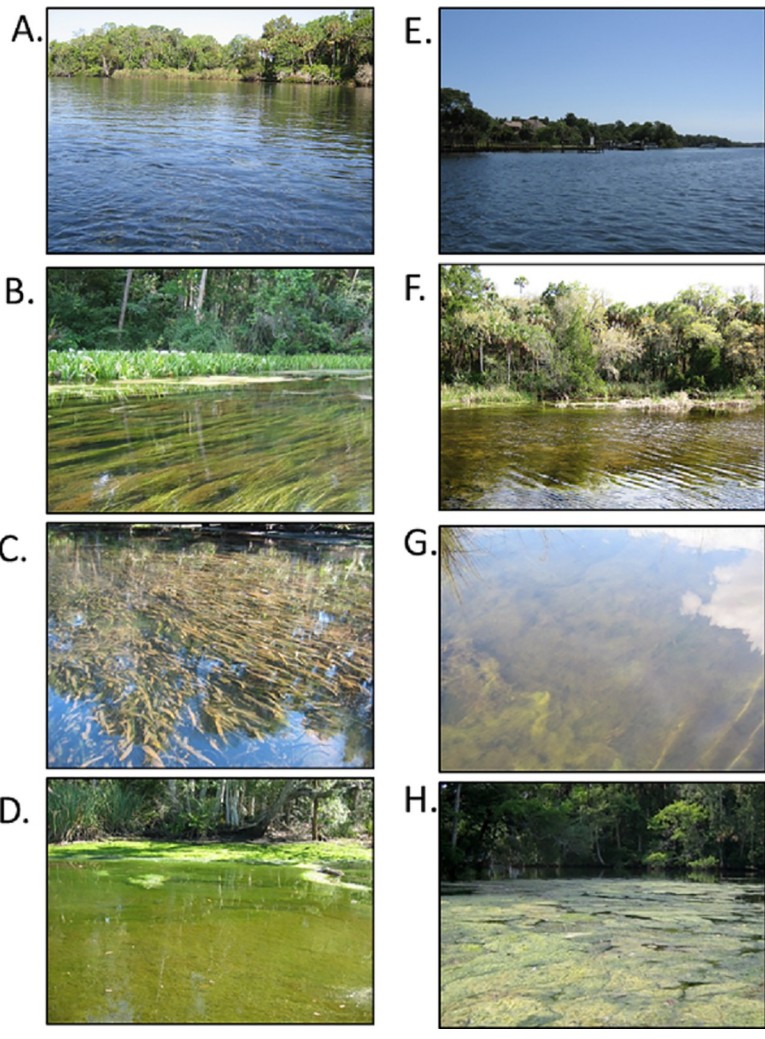

**Fig 2. Photographic comparisons of shoreline and vegetative habitats within the Chassahowitzka and Homosassa Rivers.** A) overview of the Chassahowitzka River channel and riparian habitats in Reach 1, B) mixed aquatic vegetation habitat in the Chassahowitzka River, C) dense patch of the rooted vascular plant, *Vallisneria americana*, observed during summer (July and August) in the Chassahowitzka River, D) large-scale seasonal bloom of filamentous macroalgae observed during winter (January and February) in the Chassahowitzka River, E) overview of the Homosassa River channel and riparian habitat in Reach 1, F) example of filamentous macroalgae habitat cover in the Homosassa River, G) patch of filamentous macroalgae habitat in the Homosassa River, and H) large-scale seasonal bloom of filamentous macroalgae observed during winter (January and February) in the Homosassa River.

for stomach contents were sacrificed and analyzed by dissection in the lab. All sacrificed fishes were euthanized by hypothermic shock via submersion in an ice bath. The diet composition of fishes was characterized as percent frequency by dry mass of prey groups [35, 54]. To account for partially digested and unidentifiable fish and invertebrate prey items, we multiplied the mean proportion of unidentifiable stomach contents by the percent composition of identified prey taxa, then added the partitioned unidentified mass estimates to the observed mass of each prey group, and then calculated the percent prey frequency by dry mass. Diet information and production rates for invertebrates was gathered from published literature [38–53] (Table 3). A predator-prey matrix was constructed to summarize the mean prey composition of each consumer trophic group (Table 4). To account for seasonality of migratory saltwater species foraging within the rivers, the diet composition of saltwater trophic groups was assumed to consist

**Table 1. Geographic coordinates of upstream and downstream boundaries of river reaches sampled for plants, invertebrates, and fishes in the Chassahowitzka and Homosassa rivers.** The coordinates mark the midstream locations of the reach boundaries.

| River | Reach | Midstream Boundary | Latitude | Longitude |
|---|---|---|---|---|
| Chassahowitzka | 1 | Upstream | 28.71629 | -82.57819 |
| Chassahowitzka | 1 | Downstream | 28.71648 | -82.58079 |
| Chassahowitzka | 2 | Upstream | 28.71539 | -82.58575 |
| Chassahowitzka | 2 | Downstream | 28.71671 | -82.58863 |
| Chassahowitzka | 3 | Upstream | 28.71979 | -82.60099 |
| Chassahowitzka | 3 | Downstream | 28.71527 | -82.60363 |
| Homosassa | 1 | Upstream | 28.79885 | -82.59090 |
| Homosassa | 1 | Downstream | 28.80061 | -82.59613 |
| Homosassa | 2 | Upstream | 28.79548 | -82.60343 |
| Homosassa | 2 | Downstream | 28.79253 | -82.60863 |
| Homosassa | 3 | Upstream | 28.78302 | -82.62131 |
| Homosassa | 3 | Downstream | 28.78349 | -82.62565 |

of 50% imported prey (equivalent to the approximate proportion of the year spent within the rivers versus the Gulf of Mexico [35]). This assumption was based on abundance estimates of gray snapper (*Lutjanus griseus*) and tidewater mojarra (*Eucinostomus harengulus*), the dominant saltwater large- and small-bodied fish species by biomass in each river. Additionally, terrestrial prey items (e.g., waterfowl, reptiles, terrestrial insects, and amphibians) were modeled as imported prey, similar to Gulf of Mexico imported energy sources for saltwater groups.

## Trophic mass-balance model of a coastal river food web

The Ecopath modeling framework balances the annual production in biomass of individual trophic groups with losses to predation, harvest and migration, and the overall net change in biomass. Inputs for each trophic group included the estimated biomass, production to biomass ratio, consumption to biomass ratio, prey composition, and proportion of biomass contributed to the detrital pool versus biomass exported from the system.

Using information described in the previous sections, a trophic mass-balance ecosystem model of the Chassahowitzka River aquatic food web was developed using Ecopath [55, 56, 57]. Summaries of the basic inputs for the Ecopath model are provided in Table 5. Exploitation of individual commercial and recreational species was included as part of the total instantaneous mortality rate (Z). The Ecopath model balanced when the net change in biomass within the model was accounted for by the rates of predation, harvest, and net biomass change from growth for each trophic group [58].

## Time-dynamic simulation of vascular plant loss—Effects on coastal river fish and invertebrate communities

The Ecosim framework simulates trophic group biomass rates of change over time based on gains from prey consumption times food conversion efficiency (proportion of prey consumed converted to biomass), and losses to mortality, including predation, fishing (assumed zero for the coastal river model), and unexplained natural mortality. Starting with the balanced Ecopath model, we parameterized Ecosim to assess the role of aquatic vegetation in structuring trophic interactions represented in the Ecopath model, given alternative plant-consumer mediation assumptions (Fig 3). Four specific scenarios, each representing different predator-prey dynamics coupled to vegetative habitat, were simulated using forcing functions on the

**Table 2. Trophic groups and taxa composition included in the Ecopath trophic mass-balance model of the Chassahowitzka River food web.**

| Trophic Group | Taxa Identification |
|---|---|
| Common snook | *Centropomus undecimalis* |
| Red drum | *Sciaenops ocellatus* |
| Gray snapper | *Lutjanus griseus* |
| Sheepshead | *Archosargus probatocephalus* |
| Sea catfish | *Ariopsis felis, Bagre marinus* |
| Pinfish | *Lagodon rhomboides* |
| Striped mullet | *Mugil cephalus* |
| Saltwater small-bodied fishes | *Anchoa* sp., *Gobiosoma* sp., *Leiostomus* sp., *Microgobius* sp., *Strongylura* spp., *Syngnathus* sp., *Trinectes* sp. |
| Gar | *Lepisosteus platyrhincus, Lepisosteus osseus* |
| American eel | *Anguilla rostrata* |
| Largemouth bass adults | *Micropterus salmoides* ages 1–6 |
| Largemouth bass juveniles | *Micropterus salmoides* age 0 |
| Lepomis spp. adults | *L. punctatus, L. macrochirus, L. microlophus, L. gulosus* ages 1–3 |
| Lepomis spp. juveniles | *L. punctatus, L. macrochirus, L. microlophus, L. gulosus* age 0 |
| Lake chubsucker adults | *Erimyzon sucetta* ages 1–3 |
| Lake chubsucker juveniles | *Erimyzon sucetta* age 0 |
| Freshwater small-bodied fishes | *Lucania* spp., *Menidia* sp., *Notropis* spp., *Fundulus* spp. |
| Blue crab | *Callinectes sapidus* |
| Crayfish | Cambaridae |
| Mud crabs | Grapsidae, Xanthidae |
| Grass shrimp | *Palaemonetes* spp. |
| Amphipods | *Corophium sp., Gammarus spp., Grandidierella sp., Hyalella sp.* |
| Vegetative invertebrates | Diptera larvae/pupae, Gastropoda, Isopoda, Tanaidacae |
| Benthic invertebrates | Bivalva, Nematoda, Oligochaeta, Ostracoda, Polychaeta |
| Periphyton | unknown taxa composition |
| Filamentous macroalgae | *Chaetomorpha* sp., *Gracilaria* sp., *Lyngbya* sp. |
| Vascular plants | *Vallisneria americana, Potamogeton* spp., *Hydrilla verticillata, Myriophyllum spp., Najas guadalupensis, Zanichelli* sp. |
| Sediment diatoms | unknown taxa composition |
| Detritus | unknown composition |

vascular plants and filamentous macroalgae, and predator-prey mediation functions on large- and small-bodied fishes. The forcing functions defined the magnitude and change in vegetative biomass seasonally (filamentous macroalgae) and annually over the long-term time series (vascular plants). The mediation functions modeled predator search efficiency rates as a nonlinear function of vegetative biomass (Fig 3) to simulate an increase in consumption rates of prey by predators at lower vegetative biomass.

Scenario 1 served as a baseline of plant community effects with no change in predator search efficiency. A long-term shift in river plant community structure over the 30-year period was simulated using two biomass forcing functions. The first forcing function was applied to vascular plant biomass simulating a 10-year period of constant biomass equal to the average from field observations for the Chassahowitzka River, followed by a 10-year period of linear decline, and a terminal 10-year period of vascular plant biomass equal to 50% of the initial

**Table 3. Data sources and references for the Ecopath trophic mass balance of the Chassahowitzka River food web.** In the table header, P denotes trophic group production, B denotes biomass, and Q denotes consumption.

| Trophic Group | Biomass | P/B | Q/B | Diet |
|---|---|---|---|---|
| Common snook | Capture-recapture Electrofishing [36] | Walters et al. [38] | Walters et al. [38] | Gut analysis [35] |
| Red drum | Capture-recapture Electrofishing [36] | Walters et al. [38] | Walters et al. [38] | Gut analysis [35] |
| Gray snapper | Capture-recapture Electrofishing [36] | Walters et al. [38] | Walters et al. [38] | Gut analysis [35] |
| Catfish | Depletion Seine Sampling [36] | Walters et al. [38] | Walters et al. [38] | Gut analysis [35] |
| Sheepshead | Capture-recapture Electrofishing [36] | Walters et al. [38] | Walters et al. [38] | Gut analysis [35] |
| Pinfish | Capture-recapture Electrofishing [36] | Walters et al. [38] | Walters et al. [38] | Gut analysis [35] |
| Striped mullet | Capture-recapture Electrofishing [36] | Walters et al. [38] | Walters et al. [38] | Gut analysis [35] |
| Florida gar | Capture-recapture Electrofishing [36] | Equal to 1/2 adult bass | Equal to adult bass | Gut analysis [35] |
| American eel | Capture-recapture Electrofishing [36] | Equal to adult bass | Equal to adult Lepomis spp. | Gut analysis [35] |
| Largemouth bass adults | Capture-recapture Electrofishing [36] | Estimated from growth [35] | Estimated from growth[2] | Gut analysis [35] |
| Largemouth bass juveniles | Capture-recapture Electrofishing [36] | Estimated from growth [35] | Estimated from growth[2] | Gut analysis [35] |
| Lepomis adults | Capture-recapture Electrofishing [36] | Estimated from growth [35] | Estimated from growth[2] | Gut analysis [35] |
| Lepomis juveniles | Capture-recapture Electrofishing [36] | Estimated from growth [35] | Estimated from growth[2] | Gut analysis [35] |
| Lake chubsucker adults | Capture-recapture Electrofishing [36] | Estimated from growth [35] | Estimated from growth[2] | Gut analysis [35] |
| Lake chubsucker juveniles | Capture-recapture Electrofishing [36] | Estimated from growth [35] | Estimated from growth[2] | Gut analysis [35] |
| SW small-bodied fishes | Depletion Seine Sampling [36] | Walters et al. [38] (adjusted) | Walters et al. [38] | Gut analysis [35] |
| FW small-bodied fishes | Depletion Seine Sampling [36] | Assumed equal to SWSB fishes | Assumed equal to SWSB Fishes | Gut analysis [35] |
| Blue crabs | Depletion Throw Traps [36] | Walters et al. [38] (adjusted) | Walters et al. [38] (adjusted) | Dittel et al. [42], Reichmuth et al. [43], Seitz et al. [44], Mascaro et al. [45], Rosas et al. [46] |
| Crayfish | Depletion Throw Traps [36] | Equal to blue crab (adjusted) | Equal to blue crabs | Gutierrez-Yurrita et al. [47] |
| Mud crabs | Depletion Throw Traps [36] | Equal to blue crabs (adjusted) | Equal to blue crabs (adjusted) | Kneib and Weeks [48] |
| Shrimp | Depletion Throw Traps [36] | Walters et al. [38] (adjusted) | Walters et al. [38] (adjusted) | Collins [49], Morgan [50], Costantin and Rossi [51] |
| Amphipods | Invertebrate Net Sampling [35] | Kevrekidis et al. [39], Subida et al. [40] | Equal to shrimp (adjusted) | MacNeil et al. [52], Duffy and Harvilicz [53] |
| Vegetative invertebrates | Invertebrate Net Sampling [35] | Robertson [41] (adjusted) | 2x P/B (adjusted) | Assumed 80% periphyton, 10% algae, and 10% macrophyte grazers |
| Benthic invertebrates | Sediment Cores [35] | Robertson [41] (adjusted) | 2x P/B (adjusted) | Assumed 50% detritivores/ 50% benthic grazers |
| Periphyton | Quadrat Sampling [27] | Walters et al. [38] | NA | NA |
| Filamentous Macroalgae | Quadrat Sampling [35] | Assumed equal to 10 | NA | NA |
| Vascular Plants | Quadrat Sampling [35] | Walters et al. [38] | NA | NA |
| Sediment diatoms | Frazer unpublished data | Estimated from unpublished data | NA | NA |

**Table 4. Estimated diet composition of consumers within the Chassahowitzka River, Florida.** Prey groups are listed in rows, and predator groups are listed in columns with reference numbers corresponding to the group number listed in column 1. Diet composition of fishes was based on mean percent dry mass in stomach samples taken over a three-year period. The sample sizes are shown in parentheses next to the trophic group name. The diet composition of invertebrates was based on published literature referenced in Table 3.

| Prey \ Predator | 1 | 2 | 3 | 4 | 5 | 6 | 7 | 8 | 9 | 10 | 11 | 12 | 13 | 14 | 15 | 16 | 17 | 18 | 19 | 20 | 21 | 22 | 23 | 24 |
|---|---|---|---|---|---|---|---|---|---|---|---|---|---|---|---|---|---|---|---|---|---|---|---|---|
| 1 Common snook (162) | 0 | 0 | 0 | 0 | 0 | 0 | 0 | 0 | 0 | 0 | 0 | 0 | 0 | 0 | 0 | 0 | 0 | 0 | 0 | 0 | 0 | 0 | 0 | 0 |
| 2 Red drum (56) | 0 | 0 | 0 | 0 | 0 | 0 | 0 | 0 | 0 | 0 | 0 | 0 | 0 | 0 | 0 | 0 | 0 | 0 | 0 | 0 | 0 | 0 | 0 | 0 |
| 3 Gray snapper (1,238) | 0 | 0 | 0 | 0 | 0 | 0 | 0 | 0 | 0 | 0.014 | 0 | 0 | 0 | 0 | 0 | 0 | 0 | 0 | 0 | 0 | 0 | 0 | 0 | 0 |
| 4 Catfish (15) | 0 | 0 | 0 | 0 | 0 | 0 | 0 | 0 | 0 | 0 | 0 | 0 | 0 | 0 | 0 | 0 | 0 | 0 | 0 | 0 | 0 | 0 | 0 | 0 |
| 5 Sheepshead (101) | 0 | 0 | 0 | 0 | 0 | 0 | 0 | 0 | 0 | 0.012 | 0 | 0 | 0 | 0 | 0 | 0 | 0 | 0 | 0 | 0 | 0 | 0 | 0 | 0 |
| 6 Pinfish (887) | 0.192 | 0 | 0.006 | 0 | 0 | 0 | 0 | 0.087 | 0 | 0.037 | 0 | 0 | 0 | 0 | 0 | 0 | 0 | 0 | 0 | 0 | 0 | 0 | 0 | 0 |
| 7 Striped mullet (15) | 0 | 0 | 0 | 0 | 0 | 0 | 0 | 0 | 0 | 0.004 | 0 | 0 | 0 | 0 | 0 | 0 | 0 | 0 | 0 | 0 | 0 | 0 | 0 | 0 |
| 8 Florida gar (40) | 0 | 0 | 0 | 0 | 0 | 0 | 0 | 0 | 0 | 0 | 0 | 0 | 0 | 0 | 0 | 0 | 0 | 0 | 0 | 0 | 0 | 0 | 0 | 0 |
| 9 American eel (61) | 0 | 0 | 0 | 0.079 | 0 | 0 | 0 | 0 | 0 | 0 | 0 | 0 | 0 | 0 | 0 | 0 | 0 | 0 | 0 | 0 | 0 | 0 | 0 | 0 |
| 10 Largemouth bass adults (1,002) | 0 | 0 | 0 | 0 | 0 | 0 | 0 | 0 | 0 | 0 | 0 | 0 | 0 | 0 | 0 | 0 | 0 | 0 | 0 | 0 | 0 | 0 | 0 | 0 |
| 11 Largemouth bass juveniles (366) | 0 | 0 | 0 | 0 | 0 | 0 | 0 | 0 | 0 | 0.003 | 0 | 0 | 0 | 0 | 0 | 0 | 0 | 0 | 0 | 0 | 0 | 0 | 0 | 0 |
| 12 Lepomis adults (930) | 0 | 0 | 0 | 0 | 0 | 0 | 0 | 0 | 0 | 0 | 0 | 0 | 0 | 0 | 0 | 0 | 0 | 0 | 0 | 0 | 0 | 0 | 0 | 0 |
| 13 Lepomis juveniles (844) | 0 | 0 | 0.003 | 0 | 0 | 0 | 0 | 0 | 0 | 0.046 | 0.032 | 0 | 0 | 0 | 0 | 0 | 0 | 0 | 0 | 0 | 0 | 0 | 0 | 0 |
| 14 Lake chubsucker adults (1) | 0 | 0 | 0 | 0 | 0 | 0 | 0 | 0 | 0 | 0 | 0 | 0 | 0 | 0 | 0 | 0 | 0 | 0 | 0 | 0 | 0 | 0 | 0 | 0 |
| 15 Lake chubsucker juveniles (7) | 0 | 0 | 0 | 0 | 0 | 0 | 0 | 0 | 0 | 0.003 | 0 | 0 | 0 | 0 | 0 | 0 | 0 | 0 | 0 | 0 | 0 | 0 | 0 | 0 |
| 16 SW small-bodied fishes (355) | 0.130 | 0.039 | 0.027 | 0 | 0.008 | 0.006 | 0.001 | 0.019 | 0 | 0.100 | 0.016 | 0.018 | 0 | 0 | 0 | 0 | 0.057 | 0.167 | 0 | 0 | 0 | 0 | 0 | 0 |
| 17 FW small-bodied fishes (54) | 0.099 | 0.039 | 0.051 | 0 | 0.008 | 0.010 | 0.001 | 0.680 | 0.048 | 0.374 | 0.734 | 0.018 | 0.013 | 0 | 0 | 0.010 | 0 | 0.167 | 0 | 0 | 0 | 0 | 0 | 0 |
| 18 Blue crabs | 0.004 | 0.123 | 0.010 | 0.056 | 0 | 0 | 0 | 0.083 | 0.151 | 0.014 | 0 | 0.008 | 3E-04 | 0 | 0 | 0.011 | 0.033 | 0 | 0 | 0 | 0 | 0 | 0 | 0 |
| 19 Crayfish | 0.023 | 0 | 0.040 | 0.058 | 0 | 0.021 | 0 | 0.039 | 0.383 | 0.259 | 0 | 0.032 | 0.005 | 0 | 0 | 0.004 | 0 | 0 | 0 | 0 | 0 | 0 | 0 | 0 |
| 20 Mud crabs | 0 | 0.023 | 0.033 | 0.229 | 0.016 | 0.001 | 0 | 0 | 0 | 0.006 | 0 | 0.007 | 0.001 | 0 | 0 | 0.004 | 0.023 | 0 | 0 | 0 | 0 | 0 | 0 | 0 |
| 21 Shrimp | 0.018 | 0 | 0.013 | 0.004 | 0.290 | 0.011 | 0 | 0.051 | 0 | 0.017 | 0 | 0.011 | 0.012 | 0 | 0 | 0.011 | 0 | 0 | 0 | 0 | 0 | 0 | 0 | 0 |
| 22 Amphipods | 0.011 | 0.059 | 0.279 | 0.058 | 0.142 | 0.209 | 0 | 0.006 | 0.148 | 0.054 | 0.050 | 0.633 | 0.695 | 0.113 | 0.168 | 0.221 | 0.585 | 0 | 0 | 0 | 0 | 0 | 0 | 0 |
| 23 Vegetative invertebrates | 0.023 | 0.158 | 0.028 | 0.015 | 0 | 0.127 | 0 | 0.005 | 0.058 | 0.041 | 0.067 | 0.198 | 0.219 | 0 | 0.027 | 0.074 | 0.216 | 0 | 0.400 | 0.800 | 0.300 | 0.400 | 0 | 0 |
| 24 Benthic invertebrates | 0 | 0.083 | 0.008 | 0.008 | 0.000 | 0.041 | 0.109 | 0 | 0.115 | 0.003 | 0.025 | 0.018 | 0.025 | 0.254 | 0.214 | 0.161 | 0.082 | 0 | 0.100 | 0.100 | 0.300 | 0.500 | 0 | 0 |
| 25 Periphyton | 0 | 0 | 0 | 0 | 0 | 0 | 0.334 | 0 | 0 | 0 | 0 | 0 | 0 | 0.500 | 0.500 | 0 | 0 | 0 | 0 | 0 | 0 | 0 | 0 | 0 |
| 26 Filamentous macroalgae | 0 | 0 | 0 | 0 | 0 | 0 | 0 | 0 | 0 | 0 | 0 | 0 | 0 | 0 | 0 | 0 | 0 | 0 | 0 | 0 | 0 | 0 | 0.800 | 0 |
| 27 Plants | 0 | 0 | 0 | 0 | 0 | 0.066 | 0 | 0 | 0 | 0 | 0 | 0 | 0 | 0 | 0 | 0 | 0 | 0 | 0 | 0 | 0 | 0 | 0.100 | 0 |
| 28 Sediment diatoms | 0 | 0 | 0 | 0 | 0 | 0 | 0 | 0 | 0 | 0 | 0 | 0 | 0 | 0.132 | 0.091 | 0.003 | 0 | 0.167 | 0.500 | 0.100 | 0.200 | 0.100 | 0.100 | 0.500 |
| 29 Detritus | 0 | 0 | 0 | 0 | 0 | 0 | 0.055 | 0 | 0 | 0 | 0 | 0 | 0 | 0 | 0 | 0 | 0 | 0.500 | 0 | 0 | 0.200 | 0 | 0 | 0.500 |
| 30 Marine/terrestrial import | 0.500 | 0.500 | 0.501 | 0.500 | 0.505 | 0.503 | 0.500 | 0.030 | 0.097 | 0.014 | 0 | 0.055 | 0.030 | 0 | 0 | 0.505 | 0.003 | 0 | 0 | 0 | 0 | 0 | 0 | 0 |

**Table 5. Basic inputs for the Ecopath trophic mass-balance model of the Chassahowitzka River food web.** In the table header, P/B denotes the ratio of trophic group production to biomass, and Q/B denotes the ratio of consumption to biomass.

| Trophic Group | Biomass (t/km$^2$) | P/B (annual) | Q/B (annual) |
|---|---|---|---|
| Common snook | 3.0E-04 | 0.9 | 2.4 |
| Red drum | 5.0E-05 | 1.0 | 3.0 |
| Gray snapper | 1.0E-01 | 2.0 | 20.0 |
| Catfish | 1.0E-04 | 0.8 | 7.6 |
| Sheepshead | 3.0E-04 | 2.3 | 4.0 |
| Pinfish | 6.0E-02 | 1.0 | 8.0 |
| Striped mullet | 1.4E-03 | 0.8 | 8.0 |
| Florida gar | 1.0E-05 | 0.5 | 1.0 |
| American eel | 1.0E-04 | 1.0 | 5.0 |
| Largemouth bass adults | 1.0E-02 | 1.0 | 2.0 |
| Largemouth bass juveniles | 1.2E-03 | 4.0 | 7.5 |
| Lepomis adults | 3.2E-02 | 2.0 | 5.0 |
| Lepomis juveniles | 2.4E-02 | 4.0 | 11.9 |
| Lake chubsucker adults | 3.3E-03 | 1.5 | 10.0 |
| Lake chubsucker juveniles | 2.1E-03 | 4.0 | 23.3 |
| SW small-bodied fishes | 3.2E+00 | 4.0 | 15.0 |
| FW small-bodied fishes | 1.8E+01 | 2.0 | 12.0 |
| Blue crabs | 9.2E-01 | 9.0 | 15.0 |
| Crayfish | 2.3E+00 | 1.0 | 8.5 |
| Mud crabs | 3.2E+00 | 4.0 | 9.0 |
| Shrimp | 1.5E+01 | 1.5 | 10.0 |
| Amphipods | 2.3E+01 | 7.0 | 20.0 |
| Vegetative invertebrates | 1.3E+01 | 8.0 | 20.0 |
| Benthic invertebrates | 4.0E+00 | 20.0 | 35.0 |
| Periphyton | 1.6E+02 | 25.0 | NA |
| Filamentous algae | 2.4E+02 | 10.0 | NA |
| Plants | 6.0E+02 | 9.0 | NA |
| Sediment diatoms | 1.8E+02 | 45.0 | NA |
| Detritus | 1.0E+02 | NA | NA |

biomass. A second forcing function was applied to filamentous macroalgae that simulated cyclical production occurring seasonally based on observed seasonal mean biomass estimates from vegetation monitoring in the Chassahowitzka and Homosassa rivers over the period of study (Fig 3). The plant biomass forcing functions were tuned such that model predictions matched observations from long-term field sampling [35]. Note that these forcing functions were included in all model scenarios.

Scenario 2 applied a mediation function between vascular plant biomass and the prey search efficiency of large-bodied fishes (i.e., predator search efficiency for prey increased exponentially as plant density decreased, Fig 3), thus modeling the hypothesis of compensatory predator foraging efficiency at decreased plant biomass.

In contrast to scenario 2, scenario 3 applied a mediation function between filamentous macroalgae biomass and the prey search efficiency of small-bodied fishes (i.e., predator search efficiency for prey decreased exponentially as macroalgae density increased, Fig 3). This scenario modeled the hypothesis that dense filamentous macroalgae patches served as predation refugia for macroinvertebrates.

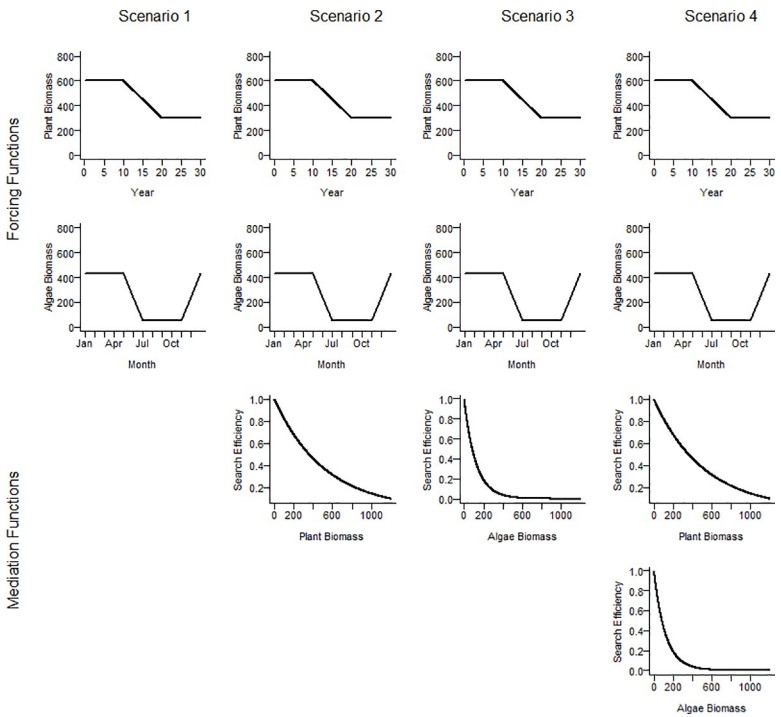

**Fig 3. Ecosim model forcing and mediation functions used to simulate changes in primary production and trophic interactions associated with the loss of aquatic vascular plants and seasonal production of filamentous macroalgae within spring-fed, coastal rivers.**

Scenario 4 included both the mediation function between vascular plants and large-bodied fishes of scenario 2 and the mediation function between filamentous macroalgae and small-bodied fishes of scenario 3 (Fig 3).

## Comparison of model accuracy and ranking of alternative models

To estimate the mean predicted biomass of trophic groups under each modeled scenario and assess model prediction error (observed vs. predicted), predicted monthly biomass for each functional group was "sampled" at the same time periods as our field sampling efforts (i.e., submersed aquatic vegetation, macroinvertebrates, and small-bodied fishes were sampled in February and August; large-bodied fishes were sampled in January and July during the terminal three years of the simulation). Similarly, model-predicted diet compositions of selected fishes were estimated from the terminal years of the time series. From these simulated time series, diet compositions were compared for the different consumers across a range of trophic levels. The predicted trophic group biomass and diet compositions under each alternative Ecosim model scenario were then compared to the observed values of the Homosassa River, and these models were ranked based on the calculated error between observed and predicted values. The total absolute error was calculated separately for biomass estimates and fish diet compositions to compare the predictive accuracy between alternative models as follows:

$$Biomass\ Error = \sum_{i=1}^{n}|obs\ B_i - pred\ B_i| \qquad (1)$$

where

$obs\ B_i$ is the observed biomass of trophic group $i$ within the Homosassa River,

$pred\ B_i$ is the Ecosim predicted biomass of trophic group $i$ under each modeled scenario,

*n* is the total number of trophic groups;
and

$$\text{Diet Composition Error} = \sum_{j=1}^{c} |obs\ DC_j - pred\ DC_j| \qquad (2)$$

*obs DC_j* is the observed diet composition of consumer *j* in the Homosassa River,
*pred DC_j* is the Ecosim predicted diet composition of consumer *j* under each modeled scenario, and
*c* is the number of consumers that were sampled for diets and occurred in both rivers.

Biomasses and diet compositions were summarized by fractional trophic levels [56], and grouped by 0.5 intervals (Table 6). This distinction was based on the dominant prey of a trophic group, and differs from traditional trophic level definitions by whole integers. The characterization was based on observed differences in prey composition among consumers within a trophic level. For example, consumers in the 2 to 2.5 trophic range had greater than 50% primary producers as prey (grazers), and consumers in the 2.5 to 3.0 trophic range had greater than 50% grazers as prey (omnivores). Similarly, consumers in the 3 to 3.5 trophic level had omnivores as a dominant prey (primary predators), and consumers within the 3.5 to 4.0 trophic level had primary predators as predominant prey type (secondary predators). These generalizations allowed for more accurate synthesis of the trophic-level effects resulting from changes in vegetation.

## Results

### Trophic groups biomasses and diet compositions from field surveys

Intensive field sampling of the Chassahowitzka and Homosassa rivers elucidated several distinct spatial and seasonal patterns (Fig 4) within the systems for filamentous macroalgae, vascular plant, invertebrate, and fish communities. High biomass of filamentous macroalgae (Genera: *Chaetomorpha*, *Chara*, *Gracilaria*, *Lyngbya*) occurred seasonally during winter in both rivers. Vascular plant (Species: *Hydrilla verticillata*, *Myriophyllum spicatum*, *Najas guadalupensis*, *Potamogeton* spp., *Ruppia maritima*, *Vallisneria americana*, *Zannichellia palustris*) biomass was higher in the Chassahowitzka River, compared to the Homosassa River where vascular plants were nearly absent (Fig 4). Macroinvertebrate biomass was high in filamentous macroalgae, with amphipods (Species: *Gammarus* spp., *Corophium louisianum*, *Grandidierella bonnieroides*, *Hyalella azteca*), gastropods (Family: Hydrobiidae), tanaids (Order: Tanaidacea), and larval insects (Orders or Suborders: Anisoptera, Coleoptera, Megaloptera, Trichoptera, Zygoptera; and Family: Chironomidae) the most common invertebrate taxa. We found larval insects and crayfish (Family: Cambaridae) commonly associated with vascular plants and in higher abundance and biomass in the Chassahowitzka River. Mud crabs (Families: Xanthidae and Grapsidae) were the only macroinvertebrate found in higher abundance and biomass in the Homosassa River across sampling events; however, invertebrates associated with

**Table 6. Definitions of trophic levels based on observed prey composition of consumers and Ecopath predicted fractional trophic level estimates.**

| Trophic level | Description | Prey Composition |
|---|---|---|
| 1 | Primary Producer | |
| 2–2.4 | Herbivore | >50% Producers, <50% Herbivores |
| 2.5–2.9 | Omnivore | <50% Producers, >50% Herbivores |
| 3–3.4 | Primary Carnivore | >50% Herbivores, <50% Omnivores |
| 3.5–4.0 | Secondary Carnivore | <50% Herbivores, >50% Omnivores, <50% Primary Carnivores |

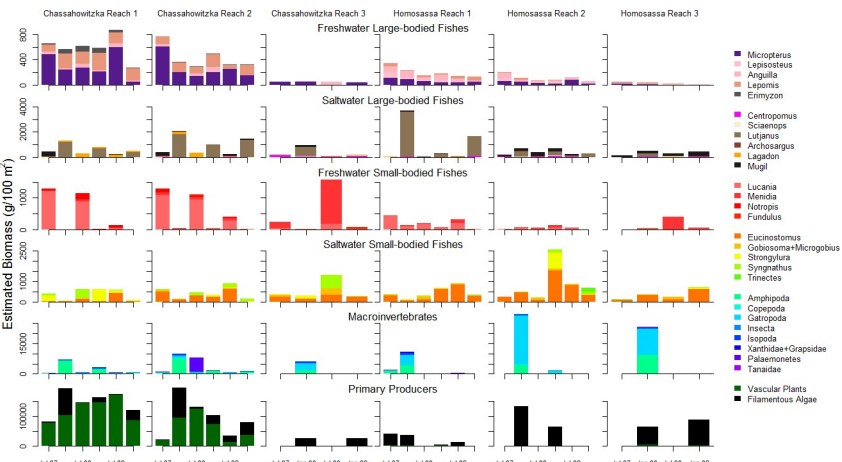

**Fig 4. Mean absolute biomass estimates of plants, macroinvertebrates, and fishes in the Chassahowitzka and Homosassa Rivers, Florida.**

filamentous macroalgae were also observed in greater biomass compared to the Chassahowitzka River during winter periods. The Chassahowitzka River generally supported a greater biomass of freshwater fishes than the Homosassa River (Fig 4). Within the Chassahowitzka River, freshwater small-bodied fishes (Genera: *Lucania* spp., *Menidia* spp., *Notropis* spp., *Fundulus* spp.) were highly abundant during summer and declined during winter. Florida gar (*Lepisosteus platyrhincus*) was the only freshwater fish species with higher abundance in the Homosassa River.

Gray snapper (*Lutjanus griseus*) were the dominant saltwater fish observed in both rivers and demonstrated seasonal patterns in abundance and biomass (Fig 4). Immigration of juvenile gray snapper (ages 0, 1, and 2) into rivers occurred during November, peak biomass was observed during December to February, and emigration to the Gulf of Mexico occurred during March to April. During periods of increased snapper density, small-bodied fish biomass decreased sharply (Fig 4), with freshwater small-bodied fishes primarily restricted to filamentous macroalgae habitat. The dominant small-bodied fish during winter periods was tidewater mojarra (*Eucinostomus harengulus*), although other saltwater species were more abundant in winter compared to summer (e.g., *Strongylura* spp.). The biomass of several large-bodied species was greater in the Homosassa River relative to the Chassahowitzka River, including common snook (*Centropomus undecimalis*), red drum (*Sciaenops ocellatus*), catfish (*Ariopsis felis*, *Bagre marinus*) and striped mullet (*Mugil cephalus*). In contrast to other saltwater fishes, pinfish (*Lagodon rhomboides*) occurred in greater biomass in the Chassahowitzka River, with peak biomass observed during summer months and in association with vascular plants. In addition to observed seasonal patterns, fish species composition in both rivers demonstrated a longitudinal gradient of highest abundance of freshwater species in reach 1 with declining abundance and replacement of freshwater by marine species downstream (Fig 4).

We documented selective foraging on prey groups that were seasonally available (detailed results in [35]). Amphipods were consumed by most fishes, including sunfish (*Lepomis* spp.), gray snapper, juvenile largemouth bass (*Micropterus salmoides*), and pinfish (Table 5). Crayfish were predominant prey for largemouth bass in the Chassahowitzka River, while largemouth bass in the Homosassa consumed a greater proportion of fishes, blue crabs, and allochthonous prey, including bull frogs (*Rana catesbeiana*) and juvenile water fowl. Fishes in the Homosassa River foraged on mud crabs more often than in the Chassahowitzka River, which was observed

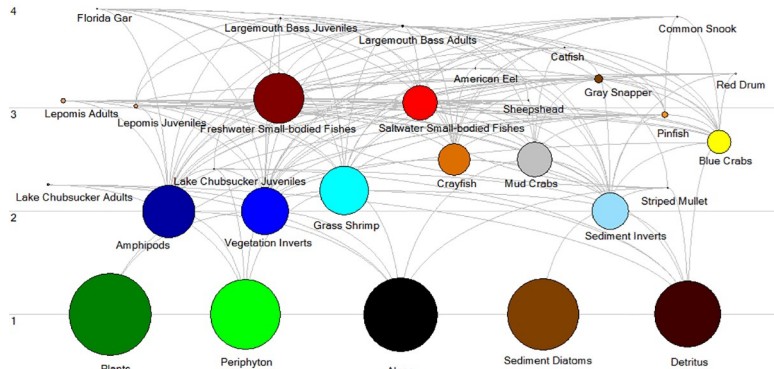

**Fig 5. Ecopath trophic flow diagram of the Chassahowitzka River food web.** The sizes of the circles represent the scale of absolute biomasses of trophic groups.

in diets of at least six species. Gray snapper diets in both rivers contained a high proportion of small-bodied fishes, including *Lucania* spp. and tidewater mojarra.

## Trophic mass-balance model

We found complex trophic interactions in the Chassahowitzka River models (Fig 5). Estimates of ecotrophic efficiency indicated high rates of transfer of invertebrate and small-bodied fish production to freshwater and marine fishes with values from 73–98% for select prey groups, including benthic invertebrates (Classes: Bivalvia, Copepoda, Nematoda, Oligochaeta, Ostracoda, Polychaeta), amphipods, blue crabs, other invertebrates associated with vegetation, and saltwater small-bodied fishes (Genera: *Anchoa mitchilli.*, *Eucinostomus* spp., *Gobiosoma* spp., *Leiostomus* spp., *Microgobius* spp., *Strongylura* spp., *Syngnathus* spp., *Trinectes* spp.). Productivity rate estimates of some invertebrate and small-bodied fish groups had to be adjusted from observed values for the Ecopath model to balance (Table 4). This was expected as some estimates of trophic group production to biomass ratios (P/B) were based on literature from systems with higher temperature fluctuations than our spring systems [7, 22].

## Predicted ecosystem responses to changes in aquatic vegetation

Time dynamic simulations of the balanced food web model predicted similar negative biomass responses of fishes and invertebrates to vascular plant loss across all scenarios (Fig 6). Scenario four was the best model in terms of biomass predictions, suggesting that key ecosystem characteristics may include: (1) bottom-up drivers of vascular plant loss and seasonal filamentous macroalgae production, (2) compensatory predator responses of increased prey search efficiency of large-bodied fishes at decreased vascular plant biomass, and (3) decreased prey search efficiency of small-bodied fishes at high filamentous macroalgae biomass (Fig 6 and Table 7). Model predictions of faunal community responses were most consistent with intersystem observed differences for large and small-bodied fishes and large crustaceans (e.g., crayfish, shrimp, blue crabs). Model predictions were less accurate for benthic and vegetation associated invertebrates (Fig 6 and Table 7). Predicted biomass declines were supported by field observations of grass shrimp (*Palaemonetes* spp.), crayfish, freshwater small-bodied fishes, lake chubsucker, pinfish, sunfish, and largemouth bass.

Ecosim-predicted diet compositions of fish consumers were similar to observed compositions (Fig 7, Table 7). Scenario three had the best predictive capability for observed diet compositions, accurately describing many changes in prey composition for predator groups and

reflecting spatial and seasonal differences in prey abundance. For example, pinfish (trophic level 2.5) diets correctly showed increased proportions of amphipods and benthic invertebrates and decreased proportions of vascular plants, while trophic level 3 consumer diets correctly showed increased proportions of saltwater small-bodied fishes and mud crabs and decreased composition of amphipods. Trophic level 3.5 consumers correctly showed increased proportions of saltwater small-bodied fishes, and consumers across all trophic levels correctly showed a decreased proportion of crayfish in diets (Fig 7). None of the models predicted the increased magnitude of mud crabs as prey composition for fishes, indicating that this prey group, along with other benthic and filamentous macroalgae associated invertebrates, may be unrecognized and important food sources for fishes in coastal rivers when vascular plants are sparse.

## Discussion

Key results from our field and model based assessments include: (1) submersed vascular plants play a central role in coastal river food webs, likely through a combination of production characteristics and mediated trophic interactions; (2) filamentous macroalgae appear to provide refuge habitat and harbors high densities and biomass of some macroinvertebrates compared to vascular plants; (3) crustaceans and small-bodied fishes in these systems provide important linkages of energy transfer from primary producers to large-bodied fishes; and, (4) the loss of vascular plants from spring-fed coastal rivers likely negatively affects the majority of invertebrates and fishes across multiple trophic levels.

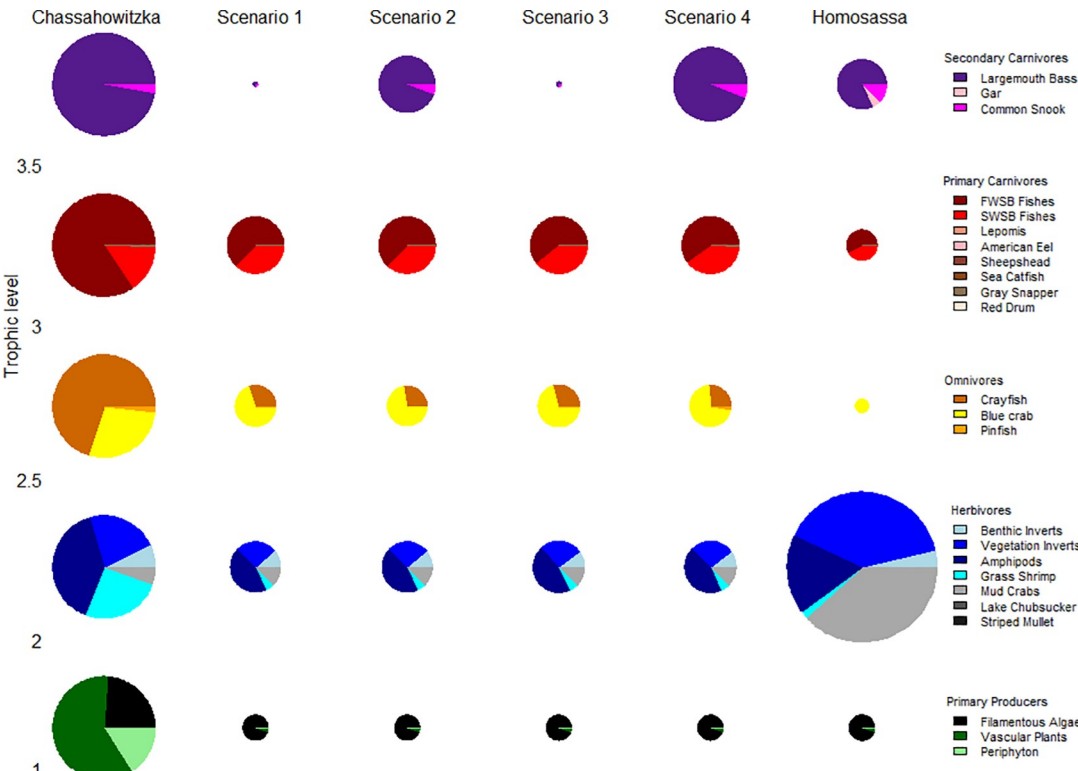

**Fig 6. Comparison of time dynamic ecosystem model predicted changes in mean biomass of trophic groups versus observed spatial differences between the Chassahowitzka and Homosassa rivers.** Note that the size of the pie charts represent absolute biomasses within a trophic level, but are scaled to the same size across trophic levels in the Chassahowitzka River. The biomass ratios of trophic levels (1 to 2 to 2.5 to 3 to 3.5) in the Chassahowitzka River was approximately 1000 to 57 to 33 to 22 to 0.01.

**Table 7. Absolute error between observed biomass and diet composition of trophic groups within the Homosassa River versus Ecosim predicted modeling of vascular plant loss and seasonal production of filamentous macroalgae under alternative hypotheses of plant mediation effects.** The values in the table are the absolute value of observed mean biomass and diet composition of fishes (measured as mean proportion dry mass of prey groups) in the Homosassa River minus the Ecopath predicted estimates.

| Trophic Group Biomass | Scenario 1 | Scenario 2 | Scenario 3 | Scenario 4 |
|---|---|---|---|---|
| Filamentous algae | 1.970 | 2.132 | 1.697 | 1.687 |
| Vascular plants | 3.094 | 3.100 | 3.082 | 3.091 |
| Periphyton | 1.756 | 1.736 | 1.910 | 1.917 |
| Benthic Invertebrates | 0.1190 | 0.1246 | 0.0295 | 0.0378 |
| Vegetative Invertebrates | 25.44 | 25.38 | 25.28 | 25.19 |
| Amphipods | 1.903 | 1.901 | 1.316 | 1.393 |
| Grass shrimp | 0.3734 | 0.3818 | 0.3147 | 0.3210 |
| Mud crabs | 28.46 | 28.46 | 28.33 | 28.33 |
| Lake chubsucker | 0.0000 | 0.0000 | 0.0000 | 0.0000 |
| Striped mullet | 0.0015 | 0.0016 | 0.0015 | 0.0014 |
| Crayfish | 0.4026 | 0.3614 | 0.4072 | 0.3480 |
| Blue crab | 0.4559 | 0.4615 | 0.4846 | 0.4972 |
| Pinfish | 0.0057 | 0.0024 | 0.0057 | 0.0292 |
| Freshwater small-bodied fishes | 3.772 | 3.759 | 3.526 | 3.457 |
| Saltwater small-bodied fishes | 1.735 | 1.717 | 2.030 | 2.031 |
| Lepomis spp. | 0.0060 | 0.0053 | 0.0067 | 0.0260 |
| American eel | 0.0000 | 0.0000 | 0.0000 | 0.0000 |
| Sheepshead | 0.0001 | 0.0001 | 0.0001 | 0.0001 |
| Sea catfishes | 0.0000 | 0.0000 | 0.0000 | 0.0001 |
| Gray snapper | 0.0493 | 0.0206 | 0.0469 | 0.0230 |
| Red drum | 0.0000 | 0.0000 | 0.0000 | 0.0000 |
| Largemouth bass | 0.0041 | 0.0014 | 0.0041 | 0.0032 |
| Florida gar | 0.0003 | 0.0003 | 0.0003 | 0.0003 |
| Common snook | 0.0006 | 0.0004 | 0.0005 | 0.0002 |
| Total Biomass Absolute Error | 69.54 | 69.54 | 68.47 | 68.39 |
| **Prey Composition of Fishes** | **Scenario 1** | **Scenario 2** | **Scenario 3** | **Scenario 4** |
| Trophic level 3.5 | 1.716 | 1.741 | 1.737 | 1.769 |
| Trophic level 3.0 | 3.669 | 3.617 | 3.645 | 3.653 |
| Trophic level 2.5 | 0.5487 | 0.6087 | 0.5512 | 0.5308 |
| Total Prey Composition Absolute Error | 5.934 | 5.966 | 5.933 | 5.953 |

Our findings indicate a possible linkage between bottom-up and top-down controls in coastal, spring-fed rivers. Plant loss is likely to affect the search efficiency of predators through decreased primary production effects on prey availability and altered predator search efficiency rates [12, 15, 59]. Our results also support the hypothesis that high plant densities adversely affects the feeding efficiency of fishes and helps sustain high densities of prey taxa [60], which may be one reason for the observed high biomass and density of select invertebrates associated with filamentous macroalgae. We found improved model predictive capability to spatially discrete biomass observations when compensatory predator search efficiencies at low vegetation biomass were included in our analysis. Modeling this response actually better explained many of the observed patterns in community structures of the coastal rivers compared to alternative models that excluded compensatory predator foraging efficiency. However, we note that our model selection approach does not account for increased model complexity due to the addition of parameters associated with the plant mediation functions.

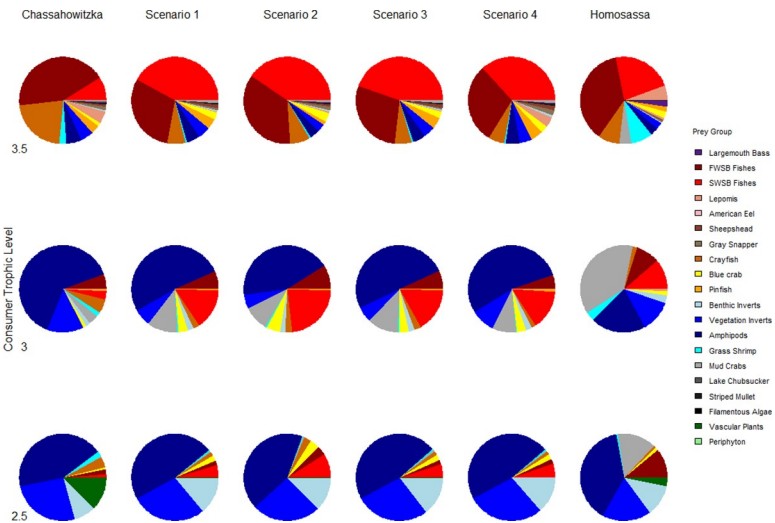

**Fig 7. Comparison of time dynamic ecosystem model predicted changes in diet compositions of select fishes versus observed spatial differences in diet composition between the Chassahowitzka and Homosassa rivers.** Pinfish comprised trophic level 2.5 consumers; *Lepomis* spp., gray snapper, red drum, and sheepshead comprised trophic level 3 consumers; and largemouth bass and common snook comprised trophic level 3.5 consumers.

We consider this limitation unimportant given the total number of parameters in the ecosystem model, because our alternative hypotheses were outlined a priori, and since our major objective was to best predict changes in river community biomass and trophic interactions associated with plant loss and replacement by filamentous macroalgae. Analyses that excluded compensatory foraging by large-bodied fishes predicted sharp population biomass declines compared to the observed biomass difference between the Chassahowitzka and Homosassa rivers. This may explain why we observed relatively low biomasses of small-bodied fishes and higher crustacean biomass in the Homosassa River–these groups may be highly susceptible to fish predators at low vascular plant biomass. Camp et al. [26] demonstrated the importance of aquatic vegetation as small-bodied fish predation refugia through tethering experiments of rainwater killifish (*Lucania parva*), concluding that predation on small-bodied fishes was greater in open substrates compared to patches of vascular plants and filamentous macroalgae within the Chassahowitzka River. Kornijów et al. [60] found that the presence of a fish predator influenced the habitat use, abundance, and size structure of *Gammarus* amphipods in a stream ecosystem, and concluded that vascular plants facilitated the coexistence of fish predator and invertebrate prey at high densities. We found similar results with many invertebrate taxa occurring in high density in plant habitats, particularly *Gammarus* spp. in filamentous algae, as well as small-bodied fishes in vascular plant colonies. In combination, the results of experimental studies in shallow aquatic ecosystems, and the findings from our field observations and ecosystem model predictions, highlight the importance of both top-down and bottom-up processes in structuring spring-fed, coastal river communities.

We hypothesized that filamentous macroalgae would mitigate predation risk to some aquatic invertebrates through reduced predator foraging rates, while at the same time competing with rooted vascular plants for light, nutrients, and other resources. We found high biomass of some macroinvertebrates in filamentous macroalgae that can be explained by decreased search efficiency of predatory small-bodied fishes. Model error was improved in terms of predicted biomass and trophic interactions of invertebrate feeders when these predator-prey interactions were included. Differences in the observed and predicted biomasses of

mud crabs and vegetation-associated invertebrates suggests these trophic groups were not accurately represented in our model. Macroinvertebrate diet habits and rates of prey consumption in these systems are poorly understood, and as a result, these trophic groups demonstrated the greatest error in predicted biomass accounting for over 77% of the total model error across all projection scenarios. This is potentially due to uncertainties in how these functional groups may respond to changes in available prey resources, and possible model misspecification of insect herbivore top-down effects on vascular plant biomass [61]. Another explanation for the discrepancy between observed and predicted small macroinvertebrate biomass is the assumption of a perfect detection in sampling. Unlike fish and larger invertebrates where detection probability was estimated [36, 37], we assumed a detection probability of 1.0 for small macroinvertebrates associated with plant habitats, which may have resulted in a negative bias in abundance and biomass estimates. Regardless of this potential negative bias, the observed responses in macroinvertebrate biomass were in stark contrast to fishes and large crustaceans, which showed a negative response to the loss of vascular plants and large-scale production of filamentous macroalgae. The various influences that affect production of invertebrates in filamentous macroalgae patches remain largely unknown and are a notable area of model uncertainty.

Empirical estimates of seasonal trophic group biomasses in each river demonstrated the dynamic community structure of these systems. During summer when vascular plant biomass was highest and filamentous macroalgae biomass was lowest, macroinvertebrate biomass was lowest, small-bodied fish biomass was highest, and large-bodied fish biomass was lowest (primarily due to emigration of marine fishes during spring). During winter, vascular plant biomass was lowest, filamentous macroalgae biomass was highest, specific invertebrate prey group (i.e., amphipods, vegetative invertebrates, mud crabs) biomasses were highest in filamentous macroalgae, small-bodied fish biomass was lowest, and the densities of marine large-bodied fish were highest (due to large-scale immigration of marine fishes in fall). For many trophic groups, model predictions matched field observations of seasonal biomass and trophic group prey compositions. Patterns in seasonal community composition were best explained by model scenario four, which incorporated compensatory search efficiency of large-bodied fishes at low vascular plant biomass during summer, and depensatory search efficiency of small-bodied fishes at high filamentous macroalgae biomass during winter.

We found that the loss of vascular plants from a coastal river ecosystem may alter the composition of the aquatic community and food web structure, and result in a net decline in the biomass of higher trophic levels, including fishes. We documented declines of specific freshwater and marine trophic groups at low vascular plant density and biomass, increased biomass of some invertebrate trophic groups associated with filamentous macroalgae biomass, and a resultant shift in fish and invertebrate faunal community composition. Our results are consistent with the findings of other researchers who have demonstrated plant loss effects on freshwater [62] and marine communities and changes in trophic dynamics [63, 64], as well as the importance of structural habitat in maintaining community structure [65]. Similar to Bettoli et al. [62], we concluded that the greatest population effects of vascular plant loss occurred for freshwater phytophilic taxa in spring-fed, coastal rivers. These large-scale ecological changes can potentially result in the loss of key species and altered food webs with implications for management of species-level biodiversity and ecosystem function [66].

The loss of grazers and detritivores from coastal rivers, including lake chubsucker and crayfish, could result in a negative feedback on vascular plant production. Lake chubsucker may utilize vegetative habitat to forage or as refugia, and were historically common in the Homosassa River when vascular plants were prevalent [21]. The loss of vascular plant habitat could have negatively affected lake chubsucker foraging success or survival and potentially led to the

extirpation of this species from portions of the river. Decline of this important herbivorous fish in the Homosassa River may have resulted in decreased grazing of periphyton, increased shading of vascular plants, and accelerated plant loss in the ecosystem [4, 67], ultimately resulting in a vegetative community comprised exclusively of seasonally abundant filamentous macroalgae. Heffernan et al. [18] proposed a similar hypothesis of decreased grazer abundance resulting in the accelerated extirpation of vascular plants within other spring-fed rivers, although the researchers propose dissolved oxygen limitation as a potential casual mechanism of grazer decline. Wood et al. [61] propose an alternative hypothesis of grazer regulated plant biomass, with systems containing low herbivore diversity experiencing higher magnitude of vascular plant loss compared to high grazer diversity systems. We measured low fish grazer diversity in both systems; however, the biomass and diversity of invertebrates classified as grazers was a key source of uncertainty in our model. While our ecosystem modelling approach and incorporation of alternative plant-consumer mediation scenarios allows for these mechanistic types of predator-prey hypotheses to be evaluated, further experimental research is needed to gain a mechanistic understanding of grazer-producer dynamics in ecosystems.

Spring discharge flow regulated river systems in Florida have been historically described as homeostatic in their chemical, physical and biological characteristics [22]. Our data suggest that the spring-fed rivers studied as part of this effort are temporally dynamic with regard to their vegetative characteristics and composition and biomass of organisms that occupy the systems. The observed differences in population densities, biomass, and diets of fishes are evidence that changes in vegetative habitat disproportionately impact individual species, and continued changes to vegetation communities are likely to alter the fish and invertebrate communities in these systems. Mechanisms leading to the loss of rooted vascular plants in Florida's springs and associated downstream waters are poorly understood; yet present restoration efforts are focused on reducing nutrient inputs and increasing flow through protection of groundwater resources and recharge areas. Our results suggest that successful plant restoration efforts are likely to result in a net increase in overall fish biomass, but that not all trophic groups will equally benefit. The complex dynamics between vascular plant biomass and consumer foraging efficiency, and filamentous macroalgae production and prey abundance, are important factors influencing the predator-prey dynamics within these systems and, in turn, the abundance and biomass of fishes and invertebrates.

## Supporting information

**S1 Zip File. Chassahowitzka River Ecopath with Ecosim model.**
(ZIP)

**S2 Zip File. Description of data files, database fields, and data codes used in the coastal river monitoring databases.**
(ZIP)

**S3 Zip File. Electrofishing catchability estimates of fish sampled at the Chassahowitzka and Homosassa rivers.**
(ZIP)

**S4 Zip File. Electrofishing sampling data collected at the Chassahowitzka and Homosassa rivers.**
(ZIP)

**S5 Zip File. Diet observations of fishes collected at the Chassahowitzka and Homosassa rivers.**
(ZIP)

**S6 Zip File. Invertebrate sampling data collected at the Chassahowitzka and Homosassa rivers.**
(ZIP)

**S7 Zip File. Submersed aquatic vegetation sampling data collected at the Chassahowitzka and Homosassa rivers.**
(ZIP)

**S8 Zip File. Compiled invertebrate databases collected at the Chassahowitzka and Homosassa rivers.**
(ZIP)

**S9 Zip File. Seine sampling data collected at the Chassahowitzka and Homosassa rivers.**
(ZIP)

**S10 Zip File. Seine captured species length-weight data collected at the Chassahowitzka and Homosassa rivers.**
(ZIP)

## Acknowledgments

We thank A. Dutterer, M. Edwards, E. Nagid, W. Strong, T. Tuten, C. Miller, A. Williams, Z. Martin, A. Cichra, and E. Buttermore for assistance with field sampling and laboratory analysis. We thank S. Sagarese and M. Karnauskas for feedback on the manuscript. We thank S. Marynowski for technical editing of the manuscript.

## Author Contributions

**Conceptualization:** Matthew V. Lauretta, William E. Pine, III, Carl J. Walters, Thomas K. Frazer.

**Data curation:** Matthew V. Lauretta.

**Formal analysis:** Matthew V. Lauretta.

**Funding acquisition:** Matthew V. Lauretta, William E. Pine, III, Thomas K. Frazer.

**Investigation:** Matthew V. Lauretta, Thomas K. Frazer.

**Methodology:** Matthew V. Lauretta, Carl J. Walters, Thomas K. Frazer.

**Project administration:** Matthew V. Lauretta, William E. Pine, III, Thomas K. Frazer.

**Resources:** William E. Pine, III, Thomas K. Frazer.

**Software:** Carl J. Walters.

**Supervision:** William E. Pine, III.

**Validation:** Matthew V. Lauretta.

**Visualization:** Matthew V. Lauretta.

**Writing – original draft:** Matthew V. Lauretta.

**Writing – review & editing:** Matthew V. Lauretta, William E. Pine, III, Thomas K. Frazer.

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
