## [Decision Letter · Decision Letter 0]

11 Sep 2019

PONE-D-19-17270

Plant mediated community structure of spring-fed, coastal rivers

PLOS ONE

Dear Dr. Lauretta,

Thank you for submitting your manuscript to PLOS ONE. After careful consideration, I feel that it has considerable merit but does not yet fully meet PLOS ONE’s publication criteria as it currently stands. Therefore, I invite you to submit a revised version of the manuscript that addresses the points raised during the review process.

Please be aware that, due to difficulty in securing a second outside reviewer, and to avoid any further delay, I chose to provide the second review by reviewing your manuscript myself. To ensure transparency, my review is clearly identified as being by me. I reviewed your manuscript prior to reading the comments by Reviewer 1, so my comments are independent of that reviewer's.

As noted above, I am asking that you prepare and submit a revised version of your manuscript. I characterized the requested revision as a major rather than minor revision because one of the recommended changes requires you to recalculate your measures of prediction error or model fit for scenarios 1 through 4.

In preparing your revision, please respond to each issue raised by Reviewer 1 and myself in our reviews. Most of these are minor issues that merely require clarification or rephrasing of various parts of the text. Three more-substantive issues I wish to highlight are the following:

1. As I explain in my review, I think it is important that you modify your measure of prediction error (or goodness of fit) to make it dimensionless, since terms in the current X^2 formula do not share the same physical dimensions and therefore should not be added. I suggest a simple alternative in my review. This change requires recalculation of the values listed in Table 6, but I will be surprised if it changes your conclusions.

2. Please clarify in the text that you are not performing statistical model selection or statistical goodness-of-fit assessments, and therefore no formal tests of statistical significance are performed. Both Reviewer 1 and I were initially puzzled by the way you currently describe your measure of goodness of fit. By calling it "chi-squared" and stating in the abstract that you perform "goodness-of-fit tests", you suggest to readers that you are performing chi-squared tests of statistical significance, but you actually do not do that.

3. Please be sure to address the question by Reviewer 1 regarding availability of all data from your study.

We would appreciate receiving your revised manuscript by Oct 26 2019 11:59PM. To enhance the reproducibility of your results, we recommend that if applicable you deposit your laboratory protocols in protocols.io, where a protocol can be assigned its own identifier (DOI) such that it can be cited independently in the future. For instructions see: http://journals.plos.org/plosone/s/submission-guidelines#loc-laboratory-protocols

A response/rebuttal letter that responds to each point raised by the academic editor and reviewer(s). This letter should be uploaded as a separate file and labeled 'Response to Reviewers'.A marked-up copy of your manuscript that highlights changes made to the original version. This file should be uploaded as a separate file and labeled 'Revised Manuscript with Track Changes'.An unmarked version of your revised paper without tracked changes. This file should be uploaded as a separate file and labeled 'Manuscript'.

Please note while forming your response: If your article is accepted, you may have the opportunity to make the peer review history publicly available. The record will include editor decision letters (with reviews) and your responses to reviewer comments. If eligible, we will contact you to opt in or out.

We look forward to receiving your revised manuscript.

Kind regards,

James N. McNair, Ph.D.

Academic Editor

PLOS ONE

Journal Requirements:

2. In your Methods section, please provide additional location information of the study area, including geographic coordinates for the data set if available.

3. To comply with PLOS ONE submissions requirements, please provide methods of sacrifice in the Methods section of your manuscript.

Additional Editor Comments (if provided):

Reviewers' comments:

Reviewer's Responses to Questions

**Comments to the Author**

1. Is the manuscript technically sound, and do the data support the conclusions?

Reviewer #1: Partly

2. Has the statistical analysis been performed appropriately and rigorously? 

Reviewer #1: Yes

3. Have the authors made all data underlying the findings in their manuscript fully available?

Reviewer #1: No

4. Is the manuscript presented in an intelligible fashion and written in standard English?

Reviewer #1: Yes

5. Review Comments to the Author

Reviewer #1: The manuscript appeared to be technically sound with the overall conclusions supported by the modeling results. Nevertheless, I did have two questions regarding the interpretation of results. (a) Is it problematic that none of the models seem to be similar to the Chassahowitka River? Why were the results of Model 1 not more similar to the Chassahowitka River (see Fig. 6)? (b) From a philosophical perspective, what if all the models you tested were "bad"? How would those results differ from what was found? The approach seems to evaluate which is the best model among the ones tested, but how do we know if any of the models are "good"?

The Ecopath/Ecosim modeling seemed appropriate as well as the approach used to evaluate competing models. I initially expected AIC to be used to evaluate models, but it is not straightforward to me how such an approach could be implemented given the structure of the models being evaluated. Some clarifications are warranted, especially for readers that do not have a deep understand of Ecopath/Ecosim. (a) P/B and Q/B ratios should be clearly defined in Table 2. Make sure Walters et al. (2008), which is cited in Table 2, is listed in the references. (b) Provide more context for interpreting the Chi-squared statistics reported in Table 6. What do small and large values indicate? What is the null hypothesis underlying the Chi-squared statistic? Is there a way to calculate p-values for the total Chi-squared statistics? How common is this type of approach (i.e., using Chi-squared statistics) for evaluating these types of models? (c) Can you provide context for why a bootstrapping procedure was used for estimating mean trophic group biomass (Lines 167-168)?

The data from the modeling output do not seem to be fully available. If this is the case, then this should be easily addressed by providing the output in supporting information.

Overall, the manuscript was well written and easy to follow. The quality of Figures 4, 6, and 7 made interpretation difficult, but this was likely the result of the file I printed.

6. PLOS authors have the option to publish the peer review history of their article (what does this mean?). If published, this will include your full peer review and any attached files.

Reviewer #1: No

---

## [Author Response · Author response to Decision Letter 0]

8 Nov 2019

All requested and suggested revisions were made to the manuscript. Please refer to the response to reviewers for detailed descriptions of the revisions and response to editor comments.

---

## [Editor Report · Decision Letter 1]

2 Dec 2019

Plant-mediated community structure of spring-fed, coastal rivers

PONE-D-19-17270R1

Dear Dr. Lauretta,

I am pleased to inform you that your manuscript has been judged scientifically suitable for publication and will be formally accepted for publication once it complies with all outstanding technical requirements.

With kind regards,

James N. McNair, Ph.D.

Academic Editor

PLOS ONE
---

## [Editor Report · Acceptance letter]

13 Dec 2019

PONE-D-19-17270R1 

Plant-mediated community structure of spring-fed, coastal rivers 

Dear Dr. Lauretta:

I am pleased to inform you that your manuscript has been deemed suitable for publication in PLOS ONE. Congratulations! Your manuscript is now with our production department. 

With kind regards,

on behalf of

Dr. James N. McNair 

Academic Editor

PLOS ONE